# Genome-Wide Identification and Salt Stress Response Analysis of the bZIP Transcription Factor Family in Sugar Beet

**DOI:** 10.3390/ijms231911573

**Published:** 2022-09-30

**Authors:** Yongyong Gong, Xin Liu, Sixue Chen, Hongli Li, Huizi Duanmu

**Affiliations:** 1Engineering Research Center of Agricultural Microbiology Technology, Ministry of Education, Heilongjiang University, Harbin 150080, China; 2Heilongjiang Provincial Key Laboratory of Ecological Restoration and Resource Utilization for Cold Region, School of Life Sciences, Heilongjiang University, Harbin 150080, China; 3Department of Biology, University of Mississippi, Oxford, MS 38677, USA

**Keywords:** sugar beet, bZIP, transcription factor, salt stress, expression pattern, bioinformatics

## Abstract

As one of the largest transcription factor families in plants, bZIP transcription factors play important regulatory roles in different biological processes, especially in the process of stress response. Salt stress inhibits the growth and yield of sugar beet. However, bZIP-related studies in sugar beet (*Beta vulgaris L.*) have not been reported. This study aimed to identify the bZIP transcription factors in sugar beet and analyze their biological functions and response patterns to salt stress. Using bioinformatics, 48 *BvbZIP* genes were identified in the genome of sugar beet, encoding 77 proteins with large structural differences. Collinearity analysis showed that three pairs of *BvbZIP* genes were fragment replication genes. The *BvbZIP* genes were grouped according to the phylogenetic tree topology and conserved structures, and the results are consistent with those reported in Arabidopsis. Under salt stress, the expression levels of most *BvbZIP* genes were decreased, and only eight genes were up-regulated. GO analysis showed that the *BvbZIP* genes were mainly negatively regulated in stress response. Protein interaction prediction showed that the *Bv*bZIP genes were mainly involved in light signaling and ABA signal transduction, and also played a certain role in stress responses. In this study, the structures and biological functions of the *BvbZIP* genes were analyzed to provide foundational data for further mechanistic studies and for facilitating the efforts toward the molecular breeding of stress-resilient sugar beet.

## 1. Introduction

Sugar beet is the main sugar crop [1] in northern China. Around the world, it also has important economic value, accounting for about 25% of the global sugar production [2]. By-products from the sugar beet pulp are rich in cellulose [3], which can be used for the preparation of material-activated carbon by adding different activators [4,5,6]. Thus, sugar beet has very good application prospects globally.

Soil salinization is a worldwide challenge [7] which threatens soil fertility and agricultural productivity [8,9]. Relevant studies have shown that under salt stress, chlorophyll biosynthesis and stomatal opening of plants are inhibited, leading to a decrease in the net photosynthetic rate and accumulation of organic matter in plants [10]. Salt stress also causes changes in membrane permeability and reactive oxygen species (ROS) production in plants [11], thus affecting the normal growth and development of plants. Studies on sugar beets showed that sugar production was significantly impaired under salt stress [12,13]. Therefore, identification and functional verification of sugar beet salt-tolerant genes are very important to provide key genetic resources for molecular breeding of sugar beets.

The basic leucine zipper (bZIP) is one of the largest transcription factor families in plants [14], and plays an important regulatory role in plant growth and development, pest defense, abiotic stress response, and other physiological processes [15,16,17,18,19]. bZIP transcription factors were first identified in the reference plant *Arabidopsis thaliana* [20,21,22] and subsequently studied in crops such as soybean (*Glycine max*) [23], rice (*Oryza sativa*) [24], maize (*Zea mays*) [25], sorghum (*Sorghum bicolor*) [26] and grape (*Vitis vinifera*) [27]. The bZIP transcription factors are named according to the bZIP conserved domain of 60–80 amino acids in length and N-X_7_-R/K-X_9_-L-X_6_-L-X_6_-L, which is composed of an alkaline DNA binding region and a Leucine zipper region [28]. The alkaline region is located at the n-terminus, and is mainly composed of Arginine and Lysine, and the nuclear localization signal is N-X_7_-R/K [29]. The Leucine zipper region has about nine amino acid residues from the C-terminal of the alkaline region, with a low degree of conservation, and one leucine residue in every seven amino acids [30].

Studies have shown that bZIP transcription factors play an important role in the plant response to salt stress [31]. In *A. thaliana*, *AtbZIP17* responds to salt stress by regulating the expression of salt-stress-responsive genes such as *AtHB-7,* thereby enhancing the salt stress tolerance of Arabidopsis [32]. Through ectopic expression of an Arabidopsis *AtbZIP60*, it was found that the superoxide dismutase activities in tobacco (*Nicotiana tabacum*) and rice (*Oryza sativa*) were significantly increased, and their tolerance to salt stress environments was concurrently enhanced [33]. Rice *OsbZIP71* regulates the levels of Na^+^ and K^+^ in the cytoplasm by binding to the promoter of an osmotic regulatory gene, *OsNHX1*, so as to reduce the concentration of Na^+^ and improve the salt tolerance [34]. In Alfalfa (*Medicago truncatula*), overexpression of *bZIP2* and *bZIP26* enhanced salt stress tolerance [35]. Heterologous expression of soybean *GmbZIP44*, *GmbZIP62* and *GmbZIP78* genes in *A. thaliana* showed that the Arabidopsis plants had better physiological indexes than the wild type under salt stress [36]. Transgenic tobacco with a *ThbZIP1* gene from *Tamarix hispida* showed accumulation of reactive oxygen species, decreased cell death and enhanced water retention, indicating the positive effect of *ThbZIP1* in plant salt tolerance [37]. Additionally, Zhang et al. [38] overexpressed a maize salt stress response gene *ABP6* in *A. thaliana*, and found that the salt stress tolerance of the transgenic Arabidopsis plants was significantly improved. Furthermore, studies in potato (*Solanum tuberosum*) have found that *StABF1* was up-regulated under salt stress [39], and overexpression of an *AREB1* enhanced tomato tolerance to salt stress [40].

The aim of this study was to identify and analyze the bZIP transcription factor family in sugar beet. The genome-wide identification of the *BvbZIP* genes was conducted using bioinformatics methods. Their gene structure, sequence characteristics, chromosome location, promoter functional elements, evolutionary relationship and salt stress response mode were analyzed, providing theoretical reference for further analysis of biological functions of the *BvbZIP* genes. The results also provide important genetic resources for molecular breeding of sugar beets for enhanced salt stress tolerance.

## 2. Results

### 2.1. Members of BvbZIP Gene Family

After verifying the structure of conserved domains, a total of 48 *bZIP* genes were identified from the whole genome of sugar beet and named on a scale from *BvbZIP1* to *BvbZIP48* according to the sequence of each gene. Expasy online programs were used to predict the sequence length, molecular weight (MW) and isoelectric point (*p*I) of the proteins encoded by the *BvbZIP* genes (Appendix A). Forty-eight *BvbZIP* genes encoded 77 proteins, among which 15 genes (*BvbZIP1*, *BvbZIP2*, *BvbZIP11*, *BvbZIP12*, *BvbZIP16*, *BvbZIP20*, *BvbZIP22, BvbZIP26, BvbZIP27, BvbZIP31, BvbZIP32, BvbZIP39*, *BvbZIP42, BvbZIP46* and *BvbZIP48*) corresponded to different transcripts and encoded multiple proteins. In addition, *BvbZIP46*, *BvbZIP47* and *BvbZIP48* genes could not be located in the beet genome due to short reads of the sequences. The sequence length of a *Bv*bZIP protein was between 141 (*Bv*bZIP28) and 841 (*Bv*bZIP10) amino acids (aas), and the molecular weight was between 16.21 kDa (*Bv*bZIP28) and 92.44 kDa (*Bv*bZIP10), and the isoelectric points were distributed between 4.78 (*Bv*bZIP7) and 9.9 (*Bv*bZIP14).

### 2.2. Sequence Characteristics of BvbZIP Gene Family

*BvbZIP* genes were divided into six groups (group1–group6) according to exon–intron structure. Except for a few genes, the *BvbZIP* gene in the same group had the same structure, with a highly similar exon number and intron phase. There were obvious structural differences between the *BvbZIP* genes in different groups. For example, the genes in group1 contained only 1 exon, while the genes in group6 contained 8 to 12 exons, reflecting a high difference between the *BvbZIP* gene sequences (Figure 1).

To further analyze the structural characteristics of *Bv*bZIP sequences, the online program MEME was used to predict 15 conserved motifs of *Bv*bZIP proteins (Figure 2a,b). Aside from the proteins encoded by the same gene, which had similar motifs, motif types and positions of other sequences were different. Analysis of the types of conserved motifs showed that almost all the sequences contained Motifs 1 and 11, which were found by multiple sequence alignment (Figure 2c). Motifs 1 and 11 corresponded to the complete bZIP domain, which is the core functional region. The topological structure of the phylogenetic tree clustered the sequences with similarly conserved motifs into one group, and there were significant differences in motifs among different groups (Figure 2a). Aside from Motifs 1 and 11, other conserved motifs only existed in part of the *Bv*bZIP sequences, such as Motifs 2, 4, 8, 13, 14, 15, etc. 

### 2.3. Chromosomal Localization and Collinearity Analysis of BvbZIP Genes

*BvbZIP46*, *BvbZIP47* and *BvbZIP48*, which had no specific chromosome location information, were excluded, and chromosome location analysis was performed on the 45 *BvbZIP* genes. Colinear genes between species were obtained by MCScanX analysis to explore whether gene duplication events existed between the *bZIP* genes. Chromosome location analysis showed that *BvbZIP* genes were distributed on nine chromosomes of sugar beet, including eight *BvbZIP* genes on chromosome 3, seven *BvbZIP* genes on chromosomes 1 and 2, six *BvbZIP* genes on chromosome 6 and five *BvbZIP* genes on chromosomes 7 and 9. Chromosomes 4, 5 and 8 contained four, two and one *BvbZIP* genes, respectively (Figure 3).

Collinearity analysis showed that there were only three pairs of segmental replication genes (*BvbZIP2* and *BvbZIP16*, *BvbZIP4* and *BvbZIP30*, *BvbZIP9* and *BvbZIP33*) between the *BvbZIP* genes in sugar beet. Comparing the collinearity of *bZIP* genes in sugar beet, Arabidopsis, rice and grape, there were 19, 24 and 13 pairs of *bZIP* genes in Arabidopsis, rice and grape, respectively, and the number of replication events of *bZIP* genes in all three species was greater than that in sugar beet (Figure 4). The results show that there were 28, 12 and 37 pairs of *bZIP* genes between sugar beet and Arabidopsis, rice and grape, respectively. There were 48 and 38 pairs of *bZIP* genes between grape, Arabidopsis and rice, respectively. There were only two pairs of *bZIP* genes between Arabidopsis and rice. In general, *bZIP* genes showed more gene duplication events between species than within species.

### 2.4. Phylogenetic Analysis of the bZIP Family

The *At*bZIP sequence and *Bv*bZIP genes were selected to construct the phylogenetic tree, which could be divided into group Ⅰ–Ⅸ according to the topological structure (Figure 5). Referring to the classification method for *At*bZIP sequences, bZIP sequences were divided into nine subclasses, A, B, C, D, E, F, G, H and S, according to the sequence structure features. BZIP sequences of the same subclass tended to cluster in the same group. Group Ⅰ was mainly composed of the D subclass bZIP sequence, including two I subclass bZIP sequences and one H subclass bZIP sequence. Group Ⅱ was composed of the S subclass bZIP, which contained one H subclass bZIP sequence. Group Ⅲ was composed of the E subclass bZIP, group Ⅳ comprised the H subclass bZIP, and group Ⅴ was composed of the B subclass bZIP. Group Ⅵ was mainly composed of A subclass bZIP sequences, including six S subclass bZIP sequences. Group Ⅶ mainly comprised the C subclass bZIP sequence, including three S subclass bZIP sequences. Group Ⅷ was composed of the G subclass bZIP sequence, and group Ⅸ comprised the F subclass bZIP sequence, including one S subclass bZIP sequence.

### 2.5. Functional Element Analysis of Promoters of BvbZIP Genes

*Cis*-acting elements play a key role in the plant response to environmental stress. The functional elements of the promoter region of *BvbZIP* genes were predicted, and a total of 10 functional elements were obtained. They were light (LRE), methyl jasmonate (MeJA), abscisic acid (ABRE), auxin (IAAR), gibberellin (GAR), anaerobic induction (ARE), low temperature (LTR), salicylic acid (SA), drought (MBS) and defense stress (TC-rich) response elements (Figure 6). *BvbZIP34* contains 36 functional elements in eight categories, while *BvbZIP31* only contains 6 functional elements in three categories. This difference may account for different biological functions of *BvbZIP* genes. All *BvbZIP* genes except *BvbZIP31* contained more than four types of functional elements, suggesting that the *BvbZIP* genes may be widely involved in the abiotic stress response of sugar beet. Among the functional elements, TC-rich, MeJA, SA, ABRE and MBS elements may be related to salt stress [41,42,43].

### 2.6. Expression Pattern Analysis of BvbZIP Genes under Salt Stress

The expression patterns of *BvbZIP* genes were analyzed using the 300 mM NaCl-treated sugar beet transcriptome data from the SRA database (Figure 7). The results show that most of the *BvbZIP* genes were down-regulated or did not change significantly after salt stress. Twelve genes showed a trend of up-regulated expression in leaves, among which *BvbZIP2*, *BvbZIP9*, *BvbZIP15*, *BvbZIP23*, *BvbZIP29* and *BvbZIP46* were significantly up-regulated. Ten genes showed a stable up-regulated expression trend in roots, among which *BvbZIP9*, *BvbZIP15*, *BvbZIP23*, *BvbZIP33*, *BvbZIP34* and *BvbZIP35* were significantly up-regulated. Comparing the expression patterns of root and leaf tissues, we found that *BvbZIP1*, *BvbZIP8*, *BvbZIP9*, *BvbZIP15*, *BvbZIP23* and *BvbZIP46* were up-regulated in both roots and leaves, suggesting that these genes play an important role in the response to salt stress in both tissues.

The transcriptomic data of sugar beet roots and leaves after 200 mM and 400 mM NaCl treatments were analyzed for the expression patterns of the *BvbZIP* genes under salt stress. A total of 45 *BvbZIP* genes were successfully matched in the transcriptome data, and the expression levels of each gene at different concentrations in the leaf (L) and root (R) were further processed. The expression level ratios of 200 mM/0 mM and 400 mM/0 mM were used to reflect the changing trends of gene expression under different concentrations of NaCl treatment. Log_2_ processing was performed for the ratio, and when the Log_2_ value was greater than 1 or less than −1, differential gene expression was indicated (Figure 8). 

In leaves, 18 *BvbZIP* genes were differentially expressed under 200 mM NaCl treatment, among which *BvbZIP16*, *BvbZIP19*, *BvbZIP29*, *BvbZIP33*, *BvbZIP34*, *BvbZIP39* and *BvbZIP43* were significantly up-regulated. Compared with untreated leaves, most *BvbZIP* genes were not differentially expressed in leaves treated with 400 mM NaCl, and only *BvbZIP29* and *BvbZIP34* were significantly up-regulated and down-regulated, respectively. In roots, the differentially expressed *BvbZIP* genes were mainly down-regulated under 200 mM NaCl treatment, and only *BvbZIP23* and *BvbZIP48* were significantly up-regulated. The expression levels of *BvbZIP16*, *BvbZIP19*, *BvbZIP29*, *BvbZIP33*, *BvbZIP34*, *BvbZIP39* and *BvbZIP43* were significantly down-regulated. There were 27 differentially expressed genes in roots under 400 mM NaCl treatment, among which 9 genes were up-regulated, *BvbZIP39* and *BvbZIP43* were significantly up-regulated, 18 genes were down-regulated, and *BvbZIP9*, *BvbZIP23* and *BvbZIP31* were significantly down-regulated.

Most of the *BvbZIP* genes showed a differential expression trend under salt stress, indicating that *BvbZIP* genes can respond to salt stress, and may play a certain role in regulating physiological activities under salt stress in sugar beet. To further quantitatively analyze the expression pattern of *BvbZIP* genes, qRT-PCR was used to detect the expression levels of the *BvbZIP* genes after 200 mM NaCl treatment for 6 h and 12 h, and no NaCl was used as the control group. After excluding the genes that could not be designed with specific primers, 29 *BvbZIP* genes were detected by qRT-PCR (Figure 9). The results show that 21 genes, including *BvbZIP4*, *BvbZIP6* and *BvbZIP7*, were down-regulated or that their expression levels did not change significantly after salt treatment; this trend is similar to that seen in previous transcriptomic data obtained in the same laboratory [44]. The expression levels of *BvbZIP3*, *BvbZIP24* and *BvbZIP44* were up-regulated 6 h after salt treatment, but down-regulated 12 h after salt treatment. The expression of *BvbZIP5* and *BvbZIP9* was up-regulated only after 12 h of salt treatment, the expression of *BvbZIP21* and *BvbZIP37* increased gradually with the increase in treatment time, and the expression of *BvbZIP43* was higher at 6 h and 12 h of salt treatment. According to the expression pattern analysis, it was determined that most members of *BvbZIP* likely play a negative regulatory role in sugar beet salt stress response.

### 2.7. Functional Annotation and Interaction Analysis of the BvbZIP Proteins

The interaction network analysis of *Bv*bZIP proteins and the selection of key modules showed that a *Bv*bZIP protein had a close interaction with other proteins involved in light and ABA signal responses (Figure 10). GO functional annotation and enrichment analysis showed that most *Bv*bZIP genes were enriched in biological processes, and the three with the highest degree of enrichment were: sucrose-induced translation repression, positive regulation of seed maturation and negative regulation of translation in response to stress (Figure 11), suggesting that *Bv*bZIP genes may be involved in the development of sugar beet seeds and negative regulation of the stress response. KEGG annotation indicates that *Bv*bZIP genes may be involved in environmental information processing and signal transduction of plant hormones (Appendix A). Among them, *Bv*bZIP3(VIP1) is involved in the cascade reaction pathway of pathogen infection and plays a regulatory role in late infection defense (Appendix A). As a downstream gene of COP1 and SPA1, *BvbZIP14*(HY5) is involved in the blue light response pathway (Appendix A). Additionally, *BvbZIP37*(ABF) regulates stomatal closure and seed dormancy in response to ABA signals (Appendix A). 

## 3. Discussion

In recent years, bZIP transcription factors have been identified and analyzed in many species, and a series of biological functions of bZIP genes in plant growth and development and abiotic stress response have been discovered. However, bZIP-related studies in sugar beet are still few. To fill this knowledge gap, this study systematically identified and analyzed bZIP transcription factors in the sugar beet genome based on bioinformatics tools and public data. A total of 48 *BvbZIP* genes were identified, which were unevenly distributed on each chromosome. According to the exon–intron structure, *BvbZIP* genes can be divided into six groups. The gene structure varies greatly among different groups, but the gene structure is relatively consistent within the same group and tends to cluster in the same branch in the phylogenetic tree. The intraspecific gene collinearity of sugar beet, grapes and *A. thaliana*, which was weaker than interspecific collinearity, suggests that *bZIP* genes in different species may not be attributed to genome replication [45], but rather that they have a common ancestor, and are distributed in different species during the process of species differentiation [46].

In addition, there were relatively few replication events between the *bZIP* genes of rice and the other three species, so it was speculated that the *bZIP* genes existed before the differentiation of monocotyledons and dicotyledons, and went through a series of replication events after differentiation, thus having higher specificity [47].

To explore the evolutionary relationship between bZIP transcription factors, the bZIP protein sequences of *A. thaliana* and sugar beet were selected to construct a phylogenetic tree, which was divided into nine groups according to the composition of the topological structure. In the study of *A. thaliana*, the bZIP transcription factors were divided into multiple a-S subclasses according to different conserved structures, among which the functions of subclasses B, E and F remain unclear [20]. Members of subclass A can respond to ABA signals, play a role in ABA response and signal transduction and may also participate in stress responses [48]. Members of subclass C have an increased number of Leucine zippers and contain nine heptapeptide repeats, which can interact with PBF protein to mediate the production of seed storage protein [49]. Members of subclass D may regulate plant development and defense mechanisms against pathogen infection [50,51]. Members of G subclass are related to ultraviolet and blue light signal transduction and participate in the regulation of light response promoters [52]. Members of subclass H contribute to the photophore formation of plants [53]. The number of members of subclass S is the largest, and the expression level of members in this group is high in vascular tissues [54]. Relevant studies have shown that some members of subclass S can be activated via transcription after stress treatment [55]. The comparison showed that the classification results of bZIP transcription factors are consistent with the topological structure of the phylogenetic tree according to the conserved structure, indicating that the members of each subclass of the *Bv*bZIP transcription factor may also have the above-mentioned different functions, and some of the differences may be caused by the existence of other motifs with low conservation.

Quantitative analysis of the salt response of the *BvbZIP* genes was carried out in beet root tissues after salt stress. The results show that *BvbZIP3*, *BvbZIP5*, *BvbZIP9*, *BvbZIP21*, *BvbZIP24*, *BvbZIP37*, *BvbZIP43* and *BvbZIP44* were up-regulated after salt stress, suggesting that these genes may play a positive regulatory role in the process of the salt response. Further ChipSeq and genetic studies may provide detailed characterization of the roles of these genes in plant salt tolerance. After salt stress, the expression of other *BvbZIP* genes was down-regulated or not changed significantly (see below). These data are consistent with the expression pattern in sugar beet transcriptome data. Compared with studies in other species, some of the *bZIP* genes were not significantly differentially expressed under short-term salt stress, but were up-regulated under long-term salt stress [56,57]. Related gene function studies also showed that some *bZIP* genes played a negative regulatory role in the plant salt stress response [58,59,60]. Combined with GO analysis results, the down-regulated *BvbZIP4*, *BvbZIP8*, *BvbZIP15*, *BvbZIP17*, *BvbZIP28*, *BvbZIP29*, *BvbZIP30*, *BvbZIP33* and *BvbZIP34* genes were enriched in the negative regulation of translation in response to stress. Other down-regulated genes are also enriched in the negative regulation items of biological processes such as development. So, it is speculated that the regulation mode of *BvbZIP* genes in salt stress is mainly negative regulation. In addition, some genes, such as *BvbZIP6*, *BvbZIP35* and *BvbZIP37*, had similar gene structures and were located in the same branch of the phylogenetic tree, but their expression patterns were greatly different, which might have been caused by different cis-acting elements.

The interaction network of *Bv*bZIP proteins revealed interested findings. For example, SPA1 protein is involved in the transmission of the plant light signal, which is related to the normal light-sensing specificity of phytochrome A [61]. BBX20 is the b-box zinc finger protein, which regulates flowering and photomorphogenesis of plants and also plays a role in the abiotic stress response [62]. FHY3 protein is a key transcription factor in phytochrome, a pathway and a positive regulator of ABA signal transduction and abiotic stress [63]. ABI3 and ABI4, ABA-insensitive transcription factors, participate in ABA signaling, reduce the sensitivity of plants to ABA, weaken the inhibition of the ABA signal in plant growth and development and play an important role in seed germination [64,65]. COP1 is involved in photomorphogenesis of plants and plays a key role in stomatal regulation and the dehydration response [66]. COP1-like regulates many developmental processes of plants, is a repressor of plant photomorphogenesis and plays an important role in regulating fruit ripening [67]. UVR8 protein can interact with COP1, and the controlled signal cascade mediates the photomorphogenesis of UV-B radiation and improves the adaptability of plants to UV radiation [68]. KEG is a Ring-E3 ligase that negatively regulates abscisic acid signal transduction [69]. The core *Bv*bZIP14 in the network is HY5, which interacts with and is degraded by COP1, the negative regulator of photomorphogenesis. COP1 is inhibited under light, and HY5 is accumulated, thus promoting photomorphogenesis in plants [70]. In conclusion, the *Bv*bZIP transcription factor interaction network mainly functions in light signal transduction, photomorphogenesis and ABA signaling pathways. Among the *Bv*bZIP genes, BBX20, FHY3 and COP1 may also be involved in the responses to abiotic stresses and dehydration. Under salt stress, photosynthesis of plants is generally inhibited [71], and ABA content is also affected. ABA plays a dual role in physiological regulation. Under salt stress, ABA accumulates to inhibit stomatal opening and plant growth [72]. During the recovery period, it is necessary to reduce the sensitivity of plants to ABA in order for the plants to grow normally [73]. Therefore, it can be reasonably expected that the *Bv*bZIP transcription factor interaction network is involved in the photosynthetic system and ABA signal transduction pathway under salt stress. The network function can improve the salt tolerance of sugar beet through regulating ABA content and organic matter accumulation.

## 4. Materials and Methods

### 4.1. Plant Material Processing

Sugar beet seeds were sterilized with thiram and germinated in vermiculite. After one week of germination, the seedlings were transferred to a hydroponic environment. The temperature of hydroponic culture was 28 °C, the intensity of light radiation was 450 μmol m^−2^ s^−1^, and the light–dark cycle was 16 h/8 h. After 4 weeks of culture, 200 mM NaCl was used as the salt stress treatment, with the control group having no addition of NaCl, and root tissues were collected after 0, 6 and 12 h of treatment [74]. The samples were stored at −80 °C for the next step of RNA extraction.

### 4.2. Gene Family Identification

Sugar beet genome, proteome data and GFF files were obtained from NCBI database (https://ncbi.nlm.nih.gov/, accessed on 20 September 2022). The bZIP domain files were obtained from Pfam database (http://pfa-m.xfam.org/, accessed on 20 September 2022) [75], and the Hidden Markov Model of the bZIP domain was constructed by the HMMER program [76]. The model was compared with sugar beet protein data, and the candidate *Bv*bZIP sequences were obtained and submitted to Pfam, SMART (https://smart.embl.de/, accessed on 20 September 2022) [77] and other online databases. The candidate sequences were further screened according to the structure of the domain.

### 4.3. Sequence Feature Analysis

The obtained protein sequences were submitted to the online database of MEME (https://meme-suite.org/, accessed on 20 September 2022) [78] to analyze the conserved motifs of *Bv*bZIP protein sequences. DNAMAN software was used to perform multi-sequence alignment to analyze the sequence composition of bZIP domain. The CDS sequence and genome sequence corresponding to *Bv*bZIP protein were extracted from GFF data and submitted to the GSDS online program (http://gsds.gao-lab.org/, accessed on 20 September 2022) [79] for exon–intron structure visualization.

### 4.4. Phylogenetic Analysis

*At*bZIP protein sequences were obtained from the *A. thaliana* genome database TAIR (https://www.arabidopsis.org/, accessed on 20 September 2022), and phylogenetic trees of *At*bZIP and *Bv*bZIP were constructed using MEGA software [80]. The construction method was the maximum likelihood method, the bootstrap value was set to 1000, and other parameters were the defaults. The ITOL (https://itol.embl.de/, accessed on 20 September 2022) online program [81] was used to further refine the build results.

### 4.5. Chromosomal Localization and Collinearity Analysis

Grape, Arabidopsis and rice were selected for collinearity analysis with sugar beet. The genome sequences and GFF files of sugar beet, grape, Arabidopsis and rice were downloaded from NCBI, TAIR and RAP-DB (https://rapdb.dna.affrc.go.jp/, accessed on 20 September 2022) databases, respectively, and the chromosome length and the location information for the *bZIP* gene on the genome of the four species were extracted. MCScanX analysis [82] was used to obtain the collinearity information between species and further analyze the gene replication events of *bZIP* gene between species. All data were visualized with TBtools software [83].

### 4.6. Promoter Analysis and Salt Stress Response Analysis

The *BvbZIP* gene CDS upstream fragment of 2000 bp was extracted as the promoter region, and the type and number of functional elements on the *BvbZIP* gene promoter were predicted using the PlantCARE database (http://bioinformatics.psb.ugent.be/webtools/-plantcare/html/, accessed on 20 September 2022) [84].The transcriptome data of sugar beet under salt stress were determined in the laboratory, other transcriptome data of sugar beet were downloaded from SRA database (Appendix A), which could be used to analyze the expression pattern of *BvbZIP* gene under 200 mM and 400 mM salt stress. TBtools was used to map gene expression.

### 4.7. cDNA Acquisition and qRT-PCR Analysis

The TRIzol method [85] was used to extract the total RNA from the roots of the sugar beet control group and salt stress treatment group, and the extracted RNA was reversely transcribed to obtain a cDNA sequence according to the instructions of the reverse transcription kit of Takara Biological Company (Dalian, China). Using the CDS sequence of *BvbZIP* genes as a template, Primer Premier 5 was used to design specific primers for qRT-PCR. The SYBR Green Ⅰ detection method and ABI Prism 7500 PCR system were used for qRT-PCR. Three biological replicates and three technical replicates were designed for each BvbZIP gene, the reference gene was 18S, and the relative expression level of the gene was calculated using the 2^−∆∆Ct^ method [86].

### 4.8. Protein Interaction Analysis and Gene Functional Annotation

*Bv*bZIP protein sequences were submitted to the String online database (https://cn.string-db.org/, accessed on 20 September 2022) to predict proteins that interact with *Bv*bZIP. Cytoscape software was used for visualization, and the MCODE plugin was used to screen key modules in the network with default parameters. Basic data were obtained from the GO Network database (http://geneontology.org/, accessed on 20 September 2022), and GO annotation, enrichment and visualization of *BvbZIP* genes were performed using TBtools. The gene ontology (GO) was divided into biological process, cell component and molecular function. KEGG annotation was carried out using KofamKOALA (https://www.genome.jp/tools/kofamkoala/, accessed on 20 September 2022). 

## Figures and Tables

**Figure 1 ijms-23-11573-f001:**
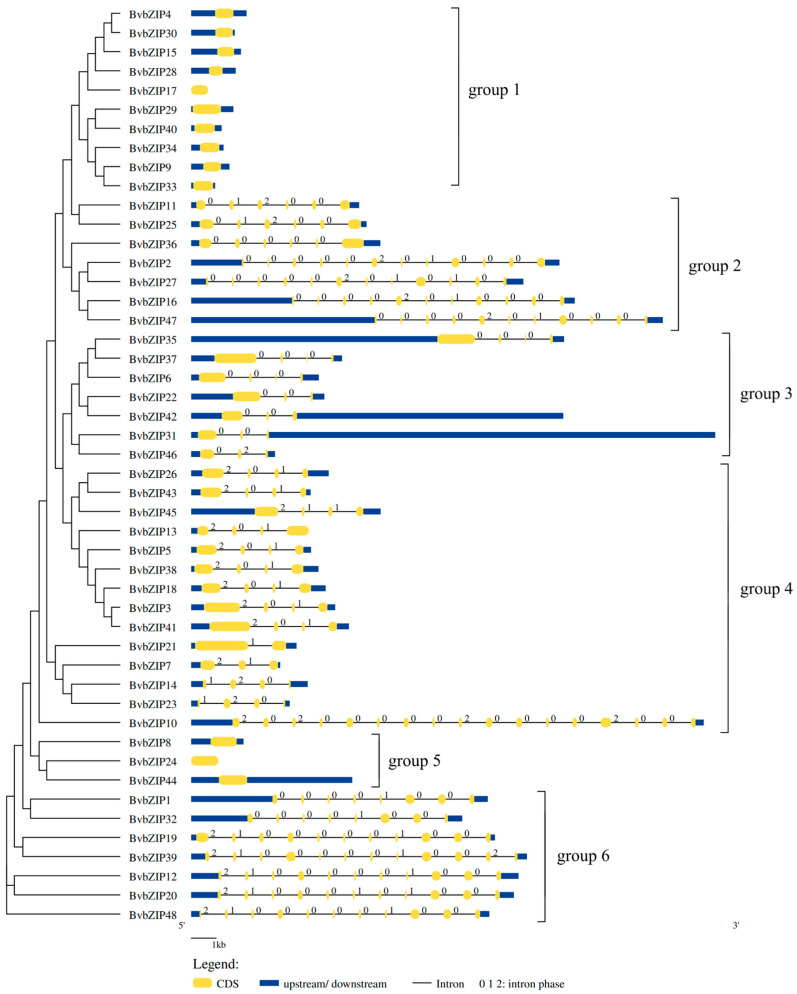
Gene structures of *BvbZIP* genes. Yellow boxes represent exons, black lines represent introns, blue boxes represent upstream and downstream noncoding regions, and the numbers 0, 1 and 2 represent intron phases. The *BvbZIP* genes were divided into groups 1 to 6 according to the intron phase, and the grouping results are consistent with the evolutionary tree structure of the *BvbZIP* genes.

**Figure 2 ijms-23-11573-f002:**
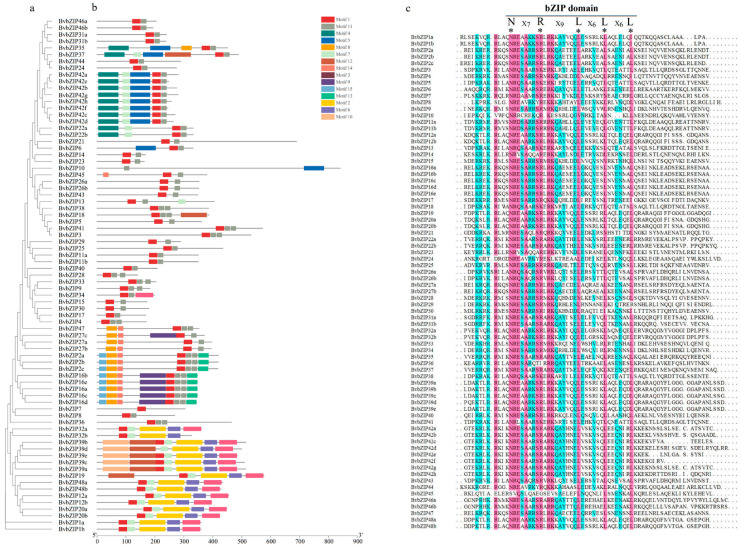
*Bv*bZIP conserved motifs and domains. (**a**) Phylogenetic tree of *Bv*bZIP proteins. (**b**) Analysis of conserved motifs of *Bv*bZIP proteins, in which motifs 1 and 11 correspond to the bZIP domain. (**c**) Multi-sequence alignment results of bZIP conserved domain, with * representing the nuclear localization signal of bZIP conserved domain.

**Figure 3 ijms-23-11573-f003:**
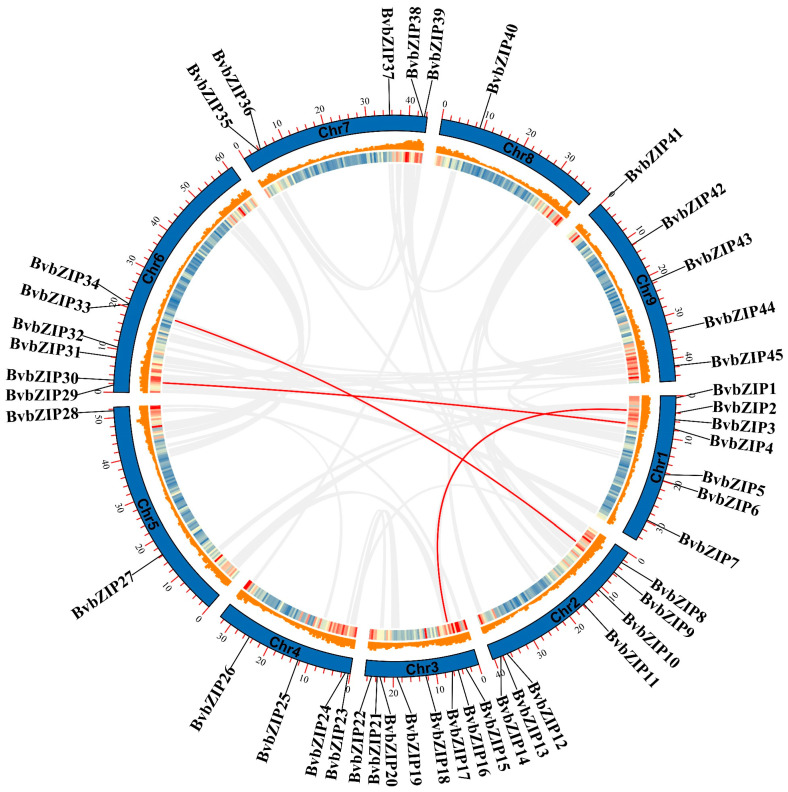
Chromosomal localization and collinearity analysis of *BvbZIP* genes. The outer circle represents the location information of *BvbZIP* genes on the chromosome, and the inner circle histogram and heat map represent the gene density and CDS density, respectively, on the sugar beet chromosome. The central line represents all fragment replication genes on the genome, and the red line connects the *BvbZIP* genes that had undergone fragment replication.

**Figure 4 ijms-23-11573-f004:**
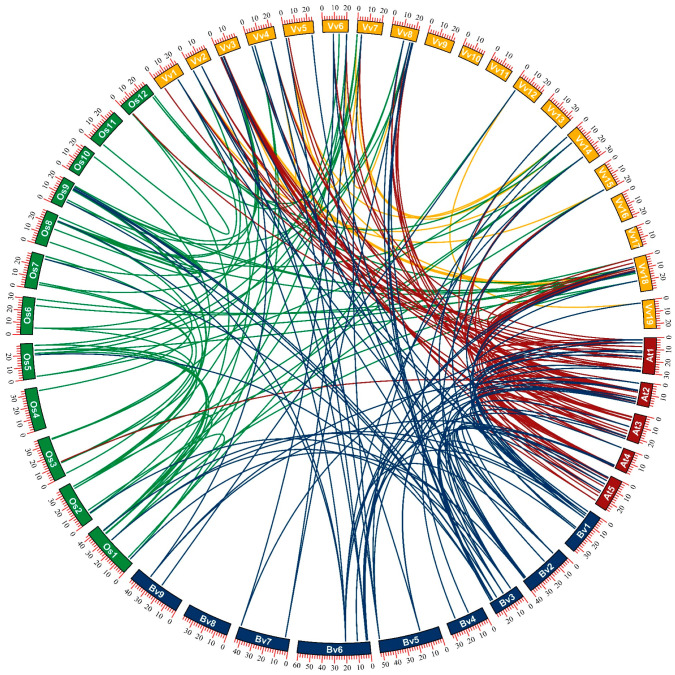
Collinearity analysis of *bZIP* genes in sugar beet, Arabidopsis, rice and grape. Blue for sugar beets, red for Arabidopsis, green for rice and purple for grapes. The lines link *bZIP* genes that were linked by fragment replication between different species.

**Figure 5 ijms-23-11573-f005:**
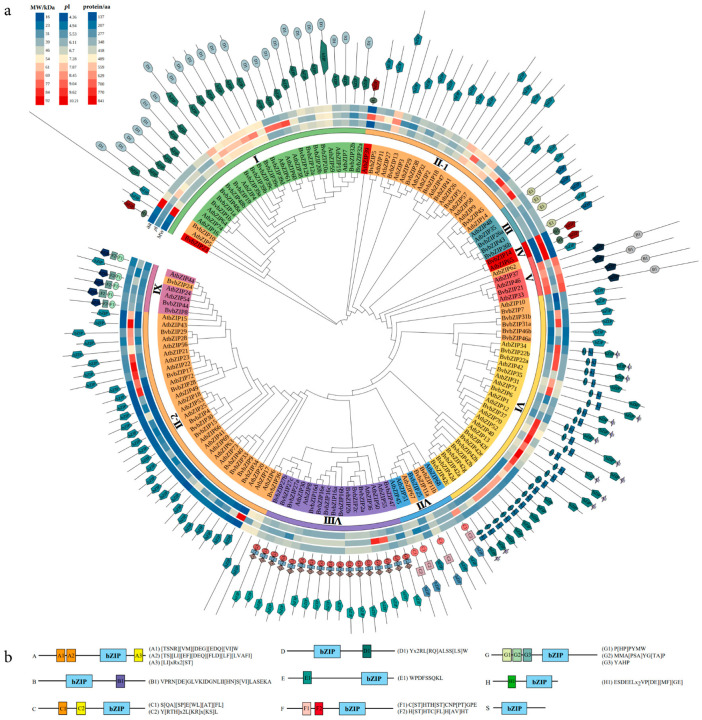
Phylogenetic analysis of the bZIP family. (**a**) Phylogenetic tree of Arabidopsis and sugar beet bZIP proteins. The outermost circle represents the conserved structural composition of each protein, and the secondary outer circle heat map represents the length, isoelectric point and molecular weight of bZIP sequence. According to the phylogenetic tree topology, bZIP was divided into groups Ⅰ–Ⅸ, and different colors of the inner circle represent bZIP genes of different subclasses A–S. (**b**) Sequence information of bZIP conserved structures in different subclasses of A–S.

**Figure 6 ijms-23-11573-f006:**
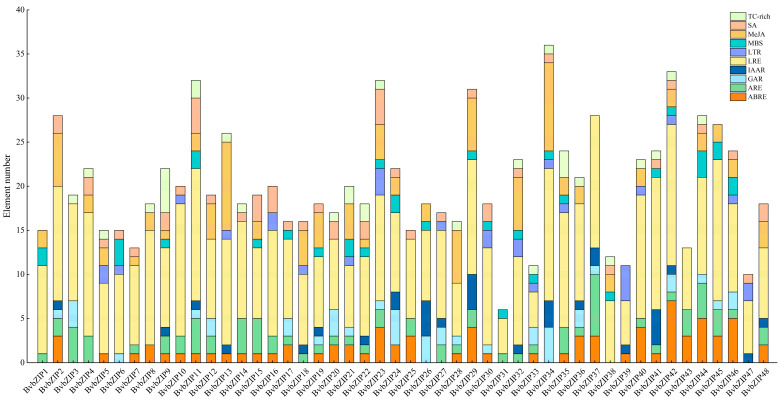
Functional element analysis of *BvbZIP* gene promoters. The vertical axis indicates the total number of functional components, the size of the box in the stack diagram indicates the number of specific functional components, and different colors indicate different types of functional components.

**Figure 7 ijms-23-11573-f007:**
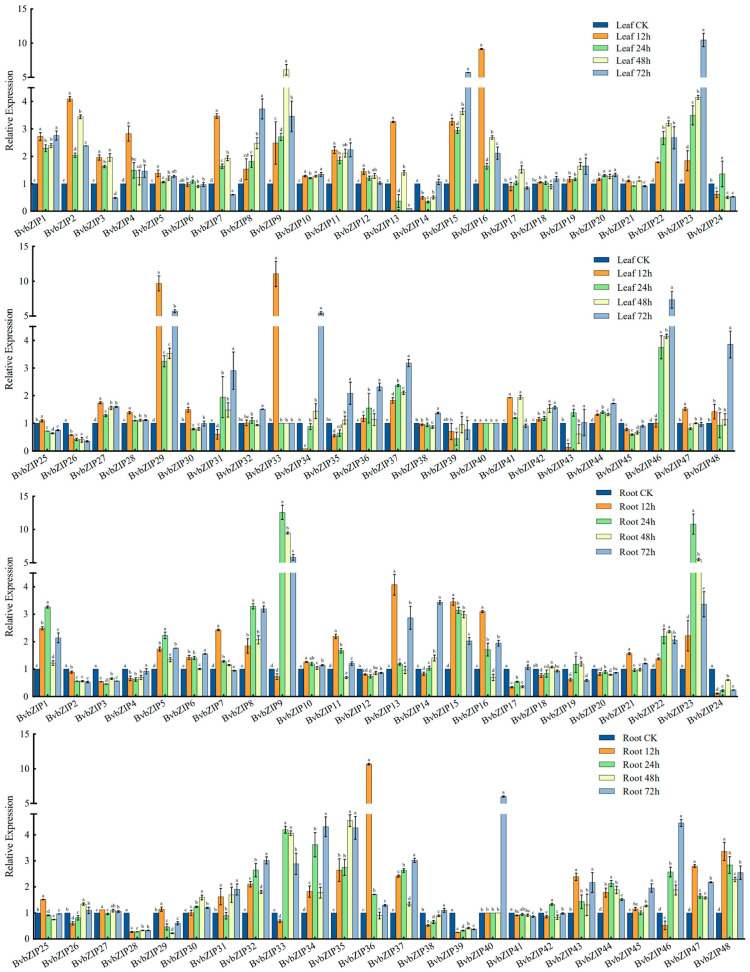
Expression pattern of *BvbZIP* genes in sugar beet roots and leaves under salt stress. Sugar beet transcriptome data after 300 mM NaCl treatment for 12 h, 24 h, 48 h and 72 h, with 0 h as the control, were obtained from the SRA database. Data were analyzed by Duncan’s analysis of variance, and different lowercase letters indicate differences in expression.

**Figure 8 ijms-23-11573-f008:**
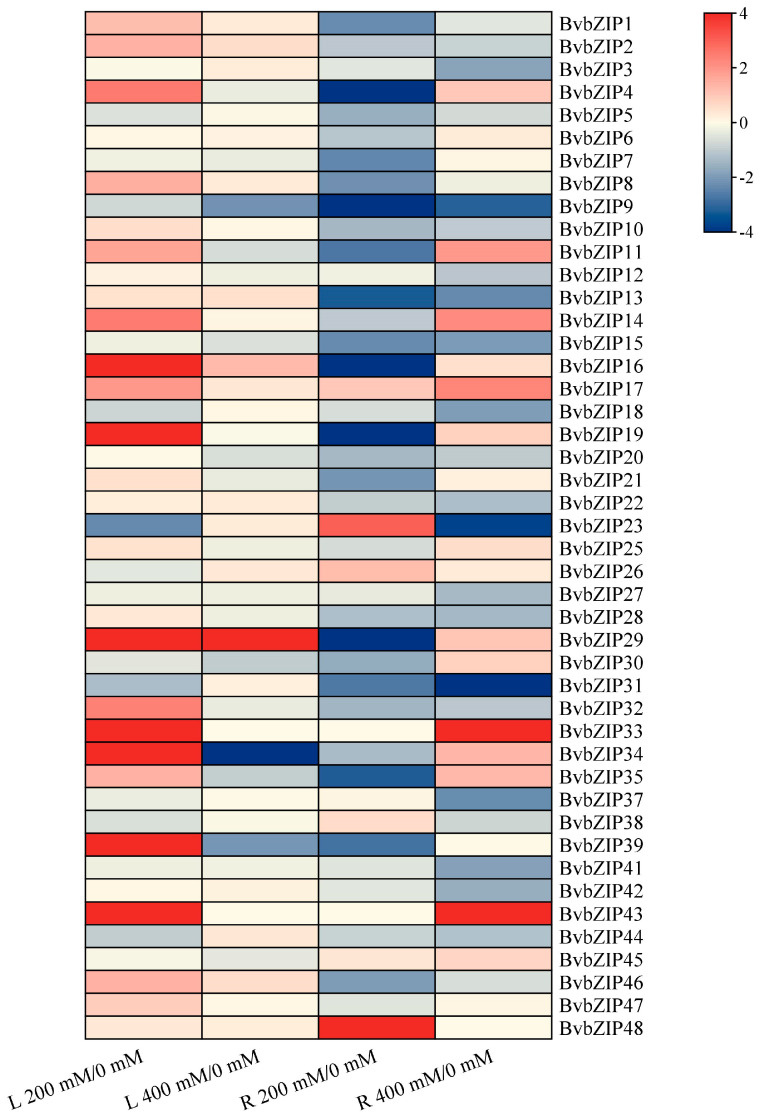
Analysis of expression patterns of the *BvbZIP* genes under salt stress. L indicates leaf tissue, R is root tissue, 0, 200 and 400 mM are different concentrations of NaCl treatment, and the expression pattern of *BvbZIP* genes is reflected by the logarithm of expression ratio under different concentrations of NaCl treatment. Blue is down-regulated expression, and red is up-regulated expression.

**Figure 9 ijms-23-11573-f009:**
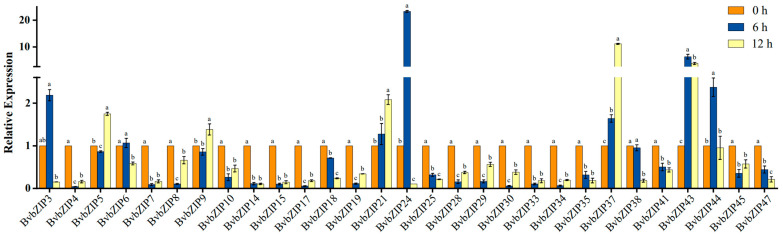
Expression patterns of the *BvbZIP* genes after salt stress. The beet root tissues treated with 200 mM NaCl were analyzed by qRT-PCR, and the sampling time was 0, 6 and 12 h. Each treatment corresponded to 3 biological replicates and 3 technical repeats. The control group was not treated with NaCl. The relative expression levels of 29 *BvbZIP* genes were calculated using the 2^−∆∆Ct^ method. The 0 h expression level was used as a reference (relative expression level was 1), less than 1 was regarded as down-regulated expression, and greater than 1 was regarded as up-regulated expression. Data were analyzed by Duncan’s analysis of variance, and different lowercase letters indicate differences in expression.

**Figure 10 ijms-23-11573-f010:**
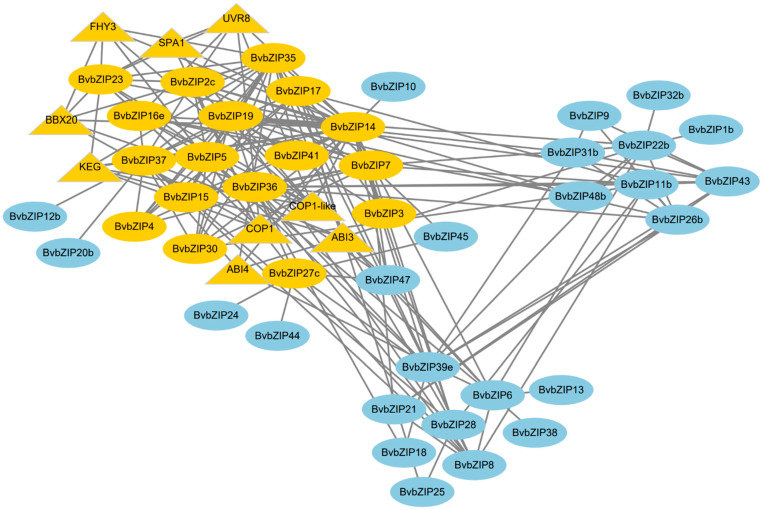
Interaction network of *Bv*bZIP proteins. The key modules of the network are colored in yellow, and triangles represent other proteins that interact with *Bv*bZIP protein.

**Figure 11 ijms-23-11573-f011:**
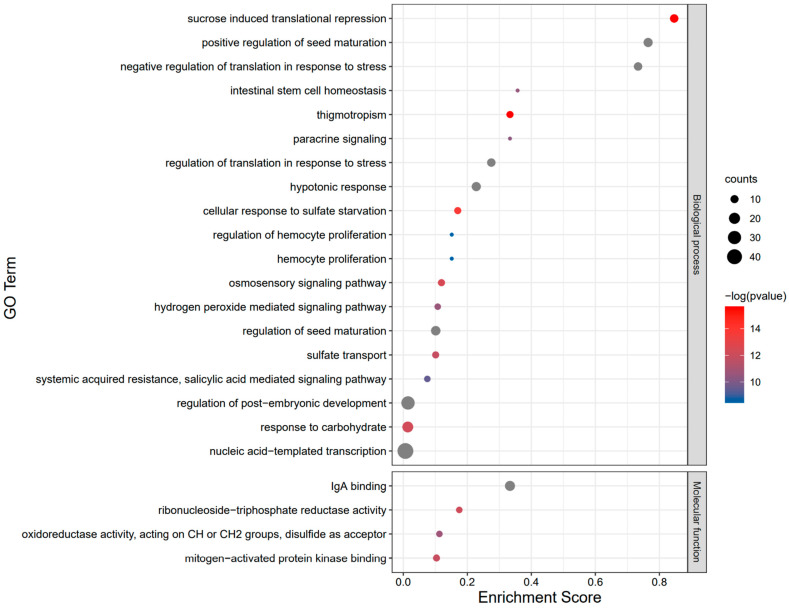
GO enrichment analysis of the *BvbZIP* genes. The horizontal axis represents the enrichment degree of genes in corresponding items, and the vertical axis shows the 23 items with the highest enrichment degree of the *BvbZIP* genes, among which the negative regulation of stress response is the key analysis item. The size of the circle represents the number of genes enriched in the corresponding project, and scale ranging from blue to red represents the *p* value from large to small, and gray represents a *p* value of 0.

## Data Availability

All the raw data, as well as gene annotations, can be found in the National Center for Biotechnology Information (https://www.ncbi.nlm.nih.gov/, accessed on 20 September 2022). All other data are available from the corresponding author upon reasonable request.

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
