# Peer review of "Genome-Wide Identification and Salt Stress Response Analysis of the bZIP Transcription Factor Family in Sugar Beet"

_ijms, 2022, doi:10.3390/ijms231911573_

Round 1

Reviewer 1 Report

The manuscript from Gong et al., represent a study on salinity response in sugar beet with focus on bZIP transcription family. A genome-wide identification has been applied to complete bZIP family in Beta vulgaris plants, and this detailed study combined bioinformatics and biological approach to reveal the genomic characteristics and potential role of BvbZIP family in salinity response. My comments on each sections was listed below.

[Abstract]

The bZIP TF family has been studied in many reference plant model, especially its contribution in numerous abiotic stress responses. Please re-considered the description on the background and purpose of studying BvbZIP family and why salinity response was selected in the first part of abstract. In addition, the scientific name of sugar beet might be spell out here or title or introduction.

[Introduction]

 1. Soil salinity is one of most serious issue worldwide and cause considerable loss for crop production, but it is not clear that whether sugar beet yield was also under the threat of salinity condition with proper references.

2. The bZIP family functioned in salinity response in reference plant species were reviewed in paragraph 4. It is not clear to reader if bZIP17 and bZIP60 were referred as example without mentioned unfolded protein response, a intracellular ER-nuclear signaling to release accumulation of unfolded protein inside of ER via transcriptional up-regulation of ER stress responsive genes. 

3. The abbreviation of scientific name for Tamarix ramosissima with Th and Lycopersicon esculentum with St  in line 68 and 74, respectively. 

[Results]

1. Line 89-91: Only eight BvbZIP genes were spelt out for their multiple transcripts and corresponding protein products among 15 genes, how these 15 were selected from 48 genes? Meanwhile, were these alternative splicing events and protein protein products produced by authors or any citation should be made?

2. Line 92 and 127: BvbZIP46-48 genes were not accurately located in the beet genome, what does it means? were they pseudo genes?

3. What is the physiological concentration of NaCl to trigger salinity stress in sugar beet and which stage of plant were used in this and also in public database? 

[Figures]

Figure 2. Please considered to change the color of motif 1 or 12 for better experience of reader to distinguish these two motifs in Fig. 2a. The motif 1 and 11 were found in the most BvbZIP genes as line 108 described, please marked the corresponding region of motif 1 and 11 separately in fig. 2c. BvbZIP7, 10 and 39d lacked of motif 1 or 11 as showed in fig. 2a, could they still functioned as bZIP TF?

Figure 4. Please considered to change the color of grape and Arabidopsis bZIP genes in collinearity analysis, the purple and red color were not easily distinguishable in the represented figure. 

Figure 6. Is there any functional element related to salinity response? Please add the title of y-axis and correct the abbreviation of salicylic acid and anaerobic induction with SA and ARE in the legend, respectively.

Figure 7. Please apply statistic analysis to the bar chart. 

Figure 8. Why the induction of BvbZIP genes under 200 mM NaCl was not shown in 400 mM NaCl group of shoot sample? Under 400 mM NaCl condition, was there any considerable symptom in shoot sample like lesion or bleaching phenotypes?

Figure 9. Which housekeeping genes did authors used to calculated relative expression of BvbZIP gene under short-term salinity treatment? Or by set 0h data as 1 as showed in current bar chart, was there statistical analysis applied to compare the BvbZIP expressions in different time point?

Figure 10. The BvbZIP protein interaction network was constructed with some Arabidopsis proteins, was it implied that any experiment was performed in current study or other references to test BvbZIP with Arabidopsis proteins? What criteria to mark the key modules as the yellow group from others?

[Discussion]

1. The bvZIP genes showed quit different expression in root and shoot tissue in public dataset and lab transcriptome in Figure 7 and 8 and later root tissue was selected to validated via RT-qPCR in Figure 9. Please elaborate the possible reasons of different behavior of BvbZIP17 among shoot and root, and why the BvbZIP expression should be quantified in root tissue under high salinity?

2. bZIP gene expression might not nicely correlate to the presence and intensity of abiotic stress as current study and listed references. For salinity response in sugar beet, do we known the downstream genes be positively and or negative regulated by bZIP family? 

3. Last paragraph was extensive discussed the interaction network of BvbZIP proteins with well-known Arabidopsis proteins functioned in light and ABA signaling. It is not clear to me how relevance it is to salinity response. Please spent some efforts on which and how bvBZIP might contribute to salinity tolerance, including pairs of bvBZIP as revealed in collinearity analysis and possible genetic modification targets in bvBZIP family to strengthen the fitness of sugar beet under unfavorable salinity environment.   

Author Response

1. [Abstract]

The bZIP TF family has been studied in many reference plant model, especially its contribution in numerous abiotic stress responses. Please re-considered the description on the background and purpose of studying BvbZIP family and why salinity response was selected in the first part of abstract. In addition, the scientific name of sugar beet might be spell out here or title or introduction.

Response: Thank you for your advice. Sugar beet is an important sugar-producing crop. As a glycophyte, it has some salt stress tolerance. Our lab is focus on studying the responses of sugar beet to salt stress. Studies of other species have shown that bZIP TF plays an important role in abiotic stresses such as drought, cold and salt. Therefore, the purpose of this study was to identify bZIP transcription factors in sugar beet and analyze their possible biological functions under salt stress. It has been modified in line 11-16 of the text. The scientific name of the sugar beet has been added to this section.

2. [Introduction]

2.1. Soil salinity is one of most serious issue worldwide and cause considerable loss for crop production, but it is not clear that whether sugar beet yield was also under the threat of salinity condition with proper references.

Response: Thank you for your advice. High soil salinity will disrupt the ion balance in the soil and cause ion damage to the roots of crops, which will further lead to osmotic stress and accumulation of ROS, resulting in crop yield reduction. The high concentration of salt affects the yield and sugar production quality of sugar beet. Relevant references have been added (corner marks 7-13).

2.2. The bZIP family functioned in salinity response in reference plant species were reviewed in paragraph 4. It is not clear to reader if bZIP17 and bZIP60 were referred as example without mentioned unfolded protein response, a intracellular ER-nuclear signaling to release accumulation of unfolded protein inside of ER via transcriptional up-regulation of ER stress responsive genes.

Response: Thank you for your advice. BZIP17 and bZIP60 can mediate the unfolded protein response, and some studies have shown that this reaction pathway can be involved in abiotic stress. However, we thought that this part of the content was not closely related to the topic, so it was not shown. Sorry for the confusion.

2.3. The abbreviation of scientific name for Tamarix ramosissima with Th and Lycopersicon esculentum with St  in line 68 and 74, respectively.

Response: Thank you for your advice. The scientific name of the species has been changed (line 68 and 74). Tamarix hispida does not have a common name, so it was not changed.

3. [Results]

3.1. Line 89-91: Only eight BvbZIP genes were spelt out for their multiple transcripts and corresponding protein products among 15 genes, how these 15 were selected from 48 genes? Meanwhile, were these alternative splicing events and protein protein products produced by authors or any citation should be made?

Response: Thank you for your questions. I'm sorry for not listing all 15 genes due to my mistake, Changes have been made (lines 92 and 93). These 15 genes were shown in the sugar beet genome annotation file to contain multiple transcripts, encoding multiple proteins. Similar results have been found in the identification of the bZIP family in other species (Yang Z, et.al. doi: 10.1186/s12863-019-0743-y).

3.2. Line 92 and 127: BvbZIP46-48 genes were not accurately located in the beet genome, what does it means? were they pseudo genes?

Response: Thank you for your questions. Due to incomplete assembly of sugar beet genome sequencing fragment, BvbZIP46-48 genes was only detected on a small fragment during sequencing, and it was not certain which chromosome it was located in. Therefore, BvbZIP46-48 genes was not a pseudo genes. This part has been rewritten for clarity.

3.3. What is the physiological concentration of NaCl to trigger salinity stress in sugar beet and which stage of plant were used in this and also in public database?

Response: Thank you for your questions. Our research group conducted a large number of proteomic studies on salt stress in sugar beet (Li H, et.al. doi: 10.1016/j.jprot.2015.03.025; Yu B, et.al. doi: 10.1016/j.jprot.2016.04.011), and determined that 400 mM NaCl was the highest NaCl concentration for maintaining the growth of sugar beet, and 200 mM NaCl was selected as the moderate salt stress, so 200 mM NaCl could induce salt stress. The data downloaded from the public database are sequencing data of four-week-old sugar beets treated with 300 mM NaCl for 0, 12, 24, 48, 72 hours.

4. [Figures]

4.1. Figure 2. Please considered to change the color of motif 1 or 12 for better experience of reader to distinguish these two motifs in Fig. 2a. The motif 1 and 11 were found in the most BvbZIP genes as line 108 described, please marked the corresponding region of motif 1 and 11 separately in fig. 2c. BvbZIP7, 10 and 39d lacked of motif 1 or 11 as showed in fig. 2a, could they still functioned as bZIP TF?

Response: Thank you for your advice. Sorry for the inconvenience caused to your review. The color of motif in the figure has been adjusted, and the corresponding positions of motif 1 and 11 are also marked in Fig. 2c. MEME website is based on sequence conservation to identify motif. Although BvbZIP7, 10 and 39d do not contain motif 1 or 11, they have bZIP domain and complete nuclear localization signal. It is possible that the domain sequence has low homology with other BvbZIP proteins, so they belong to bZIP TF.

4.2. Figure 4. Please considered to change the color of grape and Arabidopsis bZIP genes in collinearity analysis, the purple and red color were not easily distinguishable in the represented figure.

Response: Thank you for your advice. The color of grape chromosome and bZIP genes in the picture has been adjusted. Sorry for the inconvenience caused to your review.

4.3. Figure 6. Is there any functional element related to salinity response? Please add the title of y-axis and correct the abbreviation of salicylic acid and anaerobic induction with SA and ARE in the legend, respectively.

Response: Thank you for your questions and suggestions. Among the analyzed promoter elements, TC-rich, MeJA, SA, ABRE and MBS may be closely related to salt stress. Tc-rich is a defense stress response element. MBS is a drought response element, and salt stress can cause physiological water shortage in plants. MeJA, SA and ABRE elements are related to jasmonic acid, salicylic acid and abscisic acid, and can also be involved in the regulation of salt stress. Related content has been added at line 190. Title for number of elements have been added on the Y-axis, and the abbreviations for salicylic acid and anaerobic induction have also been modified.

4.4. Figure 7. Please apply statistic analysis to the bar chart.

Response: Thank you for your advice. Data have been analyzed by Duncan's analysis of variance, and different letters indicate differences in expression.

4.5. Figure 8. Why the induction of BvbZIP genes under 200 mM NaCl was not shown in 400 mM NaCl group of shoot sample? Under 400 mM NaCl condition, was there any considerable symptom in shoot sample like lesion or bleaching phenotypes?

Response: Thank you for your questions. In the early stage, our research group conducted transcriptomic studies on sugar beet salt stress. Transcriptomic measurements were carried out on sugar beet leaf and root tissues treated with 200 mM and 400 mM NaCl, and it was found that there were certain differences in gene expression patterns in different concentrations and tissues (Lv X, et.al. doi: 10.1016/j.compbiolchem.2018.04.014). Therefore, it is hypothesized that the response pattern of BvbZIP genes to 200 mM and 400 mM NaCl is different. Previous results showed that sugar beet leaves treated with 400 mM NaCl showed water-losing wilting, but no lesions.

4.6. Figure 9. Which housekeeping genes did authors used to calculated relative expression of BvbZIP gene under short-term salinity treatment? Or by set 0h data as 1 as showed in current bar chart, was there statistical analysis applied to compare the BvbZIP expressions in different time point?

Response: Thank you for your questions and suggestions. 18S RNA was used as the reference gene in the experiment, and the relative expression level of BvbZIP gene was calculated by the 2-∆∆Ct method, and the 0 h control group data was treated as 1 for data visualization. Data have been analyzed by Duncan's analysis of variance, and different letters indicate differences in expression. The corresponding content has been added at line 438.

4.7. Figure 10. The BvbZIP protein interaction network was constructed with some Arabidopsis proteins, was it implied that any experiment was performed in current study or other references to test BvbZIP with Arabidopsis proteins? What criteria to mark the key modules as the yellow group from others?

Response: Thank you for your questions. At present, there are few studies and resources  on protein function and interactions in sugar beet. Therefore, the homologous alignment method was used to predict protein interaction in this study. BvbZIP protein was compared with Arabidopsis protein in order to indirectly predict which proteins BvbZIP protein might interact with. The interaction network is visualized using Cytoscape software and the MCODE plugin is used to filter key modules, with default parameters (Degree cutoff: 2, Node score cutoff: 0.2, K-core: 2, Max. Depth: 100). The corresponding content has been added to the article at lines 443-444.

5. [Discussion]

5.1. The BvbZIP genes showed quit different expression in root and shoot tissue in public dataset and lab transcriptome in Figure 7 and 8 and later root tissue was selected to validated via RT-qPCR in Figure 9. Please elaborate the possible reasons of different behavior of BvbZIP17 among shoot and root, and why the BvbZIP expression should be quantified in root tissue under high salinity?

Response: Thank you for your question. The possible reasons for the different expression patterns of BvbZIP17 in roots and leaves are as follows: (1). Different tissues of sugar beet had different response patterns to salt stress, which was also found in previous studies (Lv X, et.al. doi: 10.1016/j.compbiolchem.2018.04.014). (2). The expression pattern of bZIP gene in different plant tissues was different, and the response pattern to abiotic stress was also different, such as the identification of bZIP in Tartary buckwheat (Liu M, et.al. doi: 10.1186/s12864-019-5882-z). Under salt stress, the roots of plants are the first to be affected, and then other tissues are affected. Therefore, the root tissues are selected for this study. According to the previous transcriptome data in the laboratory, it was found that there were more differentially expressed genes in the roots under 200 mM NaCl treatment, indicating that the bZIP gene was more responsive to medium concentration salt stress. Therefore, the root tissues under 200 mM NaCl treatment were selected for quantitative analysis.

5.2. bZIP gene expression might not nicely correlate to the presence and intensity of abiotic stress as current study and listed references. For salinity response in sugar beet, do we known the downstream genes be positively and or negative regulated by bZIP family?

Response: Thank you for your questions. In this study, it was found that the bZIP genes showed a tendency of down-regulation under abiotic stress, and similar trends were observed in other species (described in line 338 to 349). According to the predicted results of protein interaction, the downstream genes of bZIP transcription factor in sugar beet were mainly related to light signal transduction and ABA signaling pathway, and it was speculated that BvbZIP was also involved in regulating these processes. Further detailed functional studies are needed to characterize the downstream genes.

5.3. Last paragraph was extensive discussed the interaction network of BvbZIP proteins with well-known Arabidopsis proteins functioned in light and ABA signaling. It is not clear to me how relevance it is to salinity response. Please spent some efforts on which and how BvBZIP might contribute to salinity tolerance, including pairs of BvBZIP as revealed in collinearity analysis and possible genetic modification targets in BvBZIP family to strengthen the fitness of sugar beet under unfavorable salinity environment.

Response: Thank you for your questions and suggestions. The interaction network showed that BvbZIP transcription factor mainly functions in light signal transduction, light morphogenesis and ABA signaling pathways. Under salt stress, photosynthesis of plants will be inhibited and ABA content will also be affected. ABA plays a dual role in physiological regulation. Under salt stress, a large amount of ABA accumulates and inhibits stomatal opening and plant growth to maintain plant survival. During the recovery period, it is necessary to reduce the sensitivity of plants to ABA to make plants grow normally. Therefore, it is speculated that BvbZIP transcription factor interaction network is involved in the photosynthetic system and ABA signal transduction pathway under salt stress, and can improve the salt tolerance of sugar beet by regulating ABA content and organic matter accumulation. Related content has been added to this article on lines 372-383.

Reviewer 2 Report

Authors should explain in the discussion section how results from this study can be used for molecular breeding of sugar beet plants for salt tolerance.

Some references should be replaced by more recent ones (for example from 2010-2022). 

Specific comments are given in the attached file.

Author Response

1. Authors should explain in the discussion section how results from this study can be used for molecular breeding of sugar beet plants for salt tolerance.

Response: Thank you for your advice. Some BvbZIP genes were up-regulated under salt stress. Therefore, it is speculated that these genes have a positive regulatory effect on the salt tolerance of sugar beet, which can be used for further gene transformation studies to investigate their effects on the salt tolerance of plants. In addition, the predicted interaction networks may also be used to guide target selection for marker-assisted breeding. The results provide important data basis for the molecular breeding of salt tolerance varieties in sugar beet. Related content has been added to this article on lines 337-338.

2. Some references should be replaced by more recent ones (for example from 2010-2022).

Response: Thank you for your advice. In this paper, some references have been updated (in red font). Some References have not been modified because the research content is original or there has been no updates.

Round 2

Reviewer 1 Report

Dear Authors,

I have no further questions for your revision, all my comments in first round review are addressed clearly.

A minor suggestion for your future submission, please check all figures and tables for quality and resolution to publication level, and it will significantly improve your review process in my opinion.